# Perceived Stress and Impact on Role Functioning in University Students with Migraine-Like Headaches during COVID-19

**DOI:** 10.3390/ijerph20085499

**Published:** 2023-04-13

**Authors:** Nauris Tamulevicius, Ritika Save, Nishita Gandhi, Sean Lubiak, Siddhi Sharma, Claudia X. Aguado Loi, Khyam Paneru, Mary P. Martinasek

**Affiliations:** 1Department of Health Sciences and Human Performance, The University of Tampa, Tampa, FL 33606, USA; 2Department of Mathematics, The University of Tampa, Tampa, FL 33606, USA

**Keywords:** migraine-like headaches, migraine, stress, impact on role functioning, COVID-19, university students

## Abstract

Migraines, a chronic disease, can be debilitating in university students, affecting their academic performance, attendance, and social interactions. The purpose of this study was to identify the impact of COVID-19 on the role functioning and perceived stress levels of students suffering from migraine-like headaches. Methods: Two identical cross-sectional surveys were sent to students in Fall 2019 and Spring 2021 at a mid-sized university in the U.S. The students were queried on the headache impact scale (HIT-6) and perceived stress scale (PSS-10). Associations between the migraine-like headaches, severity of the headaches, stress levels, and headache impacts on the individuals’ role functioning were analyzed. Results: The average age of the respondents (n = 721) was 20.81 ± 4.32 years in 2019 and (n = 520) 20.95 ± 3.19 years in 2021. A difference (*p* = 0.044) was found in the HIT-6 score <49 category. The other categories of the HIT-6 and the PSS-10 were not significant. Conclusions: During COVID-19, more students answered that their migraine-like headaches had lower impacts on their role functioning, thus suggesting that the students were having less severe migraines. A trend was seen for student’s stress levels, indicating a decrease from 2019 to 2021. Furthermore, our results showed that the impact of headaches and stress levels slightly declined throughout the pandemic.

## 1. Introduction

Migraines are a complex, chronic disorder associated with the predominant symptoms of unilateral headache, sensitivity to light, and sound, and these symptoms are often associated with nausea and vomiting [1]. According to the headache classification committee of the International Headache Society, migraines can be divided into subtypes of being present with and without aura; the former presenting with neurological symptoms involving visual, speech, sensory, and/or motor symptoms in association with the above-mentioned predominant symptoms. [2,3]

Nearly one out of six American adults suffer from migraines, with females (20.7%) being more affected than males (9.7%) [4,5]. Compared to older adults (>50 years), younger adults (18–45 years) are more likely to experience migraine symptoms (25.5%) [4,5]. Young adults are exposed to migraine-triggering factors such as stress, anxiety, lack of sleep, increased screen time, and fatigue [6]. Migraines can be debilitating in university students, affecting their academic performance, attendance, and social interactions in school [7,8,9]. Students with migraines report a higher prevalence of depression, anxiety, reduced productivity at work/home, and frequent visits to a medical provider [8,9].

There has been more research on individuals suffering from migraines after the emergence of the SARS-CoV-2 virus or as a response to infection [10]. Overall, the pandemic has had negative impacts on the quality of life and role functioning of people suffering from chronic diseases, such as heart disease, diabetes, cancer, chronic obstructive pulmonary disease, chronic kidney disease, etc. [11]. The long-term effects on individuals infected with COVID-19 have included prolonged migraine-like headaches which last for months despite recovery from COVID-19 infection [12]. It has been speculated that patients suffering from migraines are more susceptible to contracting COVID-19 due to shared inflammatory and cardiovascular mechanisms [13]. The negative impacts of COVID-19 infection, lockdown and isolation, and manifestations of migraines have had a cascading series of negative health events. Social distancing protocols have negatively impacted individuals with chronic migraines because of limited outdoor activities, which directly affects sleep quality and subsequently results in an increased frequency of headaches and worsening in the clinical course of migraines. This is likely due to the increased psychological and emotional stresses associated with lockdowns [14,15]. Additionally, an increase in the frequency, severity, and chronicity of migraine attacks after lockdowns has resulted in increased anxiety and depression [16]. Conversely, there are studies that have shown that lockdowns had more positive effects on individuals suffering from migraines due to decreases in work stress [17,18,19], reduced social burdens [17], and a general sense of well-being [17]. In the adolescent population, reductions in migraines have been attributed to decreased school-related stressors [20,21]. To the best of our knowledge, there have been no studies assessing the effects of the COVID-19 outbreak and lockdowns on university/college students. Our study aimed to identify the impact of COVID-19 on the role functioning and perceived stress levels of students suffering from migraine-like headaches.

## 2. Materials and Methods

A cross-sectional survey using the Qualtrics online platform was conducted at a mid-sized university (approximately 10,000 students) in October 2019 and in March 2021. This was used to identify the impacts of the COVID-19 outbreak and lockdowns on the severity, frequency, and duration of migraine-like headaches in students. The survey and electronic consent form were sent to graduate and undergraduate students via an email link. The inclusion criteria were students 18–40 years of age and those currently attending the university.

At least one migraine-like headache in the past six months was an inclusion criterion which was determined by a qualifying self-report question at the beginning of the survey asking if they had experienced migraine-like headaches. Those who responded ‘No’ to this question were not included in further analyses. The questionnaire comprised two parts in 2019: (I) the first part aimed to gather data pertaining to demographics and the severity, frequency, and duration of migraine-like headaches; and (II) the second part analyzed the impacts of headaches on quality of life and perceived stress levels using the following pre-validated tools: the Headache Impact Test-6 (version 1.1) and the Perceived Stress Scale-10 (PSS-10), respectively [22,23]. In 2021, a third part was added to the questionnaire, in addition to the above two parts. The third part focused on self-developed questions pertaining to COVID-19 infection, lockdown, and the participant’s perceived disabilities with respect to migraine-like headaches.

The HIT-6 tool (Cronbach’s alpha = 0.87) is comprised six questions that aim to identify the severity of headaches and the social and emotional impacts of these headaches on an individual’s role functioning [22,24]. The responses are in the form of a Likert scale ranging from 6 (never) to 13 (always) [24]. Thus, the total scores can range from 36 to 78, with higher scores indicating a greater impact [24]. Scores of ≤49 represent little or no impact, scores of between 50–55 suggest some impact, scores of between 56–59 represent a substantial impact, and scores of greater than 60 suggest a severe impact on role functioning [22,24]. This tool has high validity and reliability in predicting the severity of migraine headaches in a clinical population [25,26].

The PSS-10 scale comprises ten questions used to assess stress levels in adults above 12 years of age [23,27]. The responses range from 0 (never) to 4 (very often) on a Likert Scale to analyze subjective stress levels in participants. The total scores are calculated by adding up the responses for each question, and they can vary from 0 to 40, with higher scores indicating higher stress levels [28]. For clinical research, the severity of stress can be categorized into 3 groups based on the PSS scores: low stress (0–13), moderate stress (15–27), and high stress (27–40) [29]. The PSS-10 is a valid and reliable tool for measuring perceived stress levels in English-speaking populations [30].

Chi-square (*χ*2) tests were conducted to assess associations between the HIT-6 and COVID-19 infection as well between the PSS-10 and COVID-19 infection. MS-Excel (O365) was used to tabulate the data and calculate the number of participants in each score category. SPSS software (version v.27) was used to conduct the Chi-square (*χ*2) test. R Statistical Software (v4.1.2, 2021) was used to compare intragroup differences in each score category of the HIT-6 and PSS-10 scores between the two cohorts (2019 and 2021). 

## 3. Results

The survey response rates were 7.6% (721 out of 9535) in 2019 and 5.6% (520 out of 9304) in 2021. The average age of respondents was 20.87 ± 3.89 for both years (Table 1).

The demographics for both samples are presented in Table 2 and Table 3. During both survey administration times, 90.01% and 88.65% of the respondents in the first and second cohort, respectively, were undergraduate students, with most of the participants (40.36% and 45.19%, respectively) primarily in the middle of their program of study. In 2021, the additional COVID-19 questionnaire included questions about the COVID-19 outbreak and infection.

Of those students in the COVID cohort, 21.9% had tested positive for COVID-19 in the past 12 months. Out of those who tested positive, 43.6% of them had mild symptoms and 29.1% had moderate symptoms. Amongst those infected, 47.3% reported no difference in their migraine-like headaches. Additionally, 18.2% reported an increase in the frequency of their migraine-like headaches, whereas 9.1% reported an increase in the intensity, frequency, and duration of the headaches after being infected with COVID-19. The majority of those infected (65.4%) had no side effects and had fully recovered. However, only 1.8% reported severe side effects that affected their daily activities.

The results indicated that 53.4% of the individuals reported no difference in their migraine-like headaches due to the COVID-19 outbreak. However, 21.1% of the students reported an increase in the frequency of their migraine-like headaches after the COVID-19 outbreak.

### 3.1. HIT-6 Questionnaire

In 2019, 96.47% (n = 453) of the sample, and in 2021, 94.52% (n = 292) of the sample, reported scores of greater than 50 for the HIT-6 questionnaire. Our results indicated that 11.9% of the students had scored between 50 and 55, which represents some impact on their role functioning in 2019, whereas this percentage increased to 12.84% in 2021. Conversely, a decrease was seen in these percentages, with 15.23% of the students scoring in the range of 56 to 59, which indicates an impact on the role functioning, to only 13.51% of the students scoring in the same range. A similar trend was seen in these percentages for the years 2019 and 2021 wherein 69.98% of the students scored above 60 on the HIT-6 questionnaire, which suggested a severe impact on their role functioning; however, this percentage fell to 66.89% in 2021. When assessing the intragroup difference between the HIT-6 scores for the two cohorts, there was a significant increase in the number of participants reporting scores of ‘<49’ in 2021 (*p* = 0.044). The remaining score categories did not have any significant differences between the two years (Table 4).

### 3.2. PSS-10 

In 2019, 636 students responded to the PSS-10 questionnaire, whereas, in 2021, 461 students responded. In 2019, 66.35% of the students, and in 2021, 68.55% of the students, had scores between 27 and 40 (severe impact of stress) on the PSS questionnaire. In 2019, 33.33% of the students, and in 2021, 30.37% of the students, had scores between 14 and 26 (moderate impact of stress) on the PSS-10 questionnaire. There was no significant difference (*p* > 0.05) in any of the score categories for the PSS-10 between the two years (Table 4).

A chi square test was conducted to assess the severity of migraine-like headaches in students with or without COVID-19. Pearson’s chi square value was 3.970 with a *p*-value of 0.265 (HIT-6).

Similarly, a chi-square test was conducted to identify the stress impact on students with or without migraine-like headaches. The obtained Pearson’s chi-square value was 0.392, with a *p*-value of 0.822 (for the PSS-10).

## 4. Discussion

This study focused on the impacts of COVID-19 on role functioning and perceived stress for migraine-like headaches in college students. There was a significant difference in the HIT-6 score <49 category (*p* = 0.044). The other HIT-6 categories indicated that a lower number of university students suffered from migraine-like headaches, although it was not significant. Our results are inconsistent with past studies which show that migraines or migraine-like headaches have a strong association with increased stress [27,30,31,32] and poor role functioning [27,33,34,35,36]. Additionally, there are past studies that have indicated increased new-onset headaches and increased stress disturbances post-COVID-19 [37].

A small decrease, though not statistically significant, was observed in the PSS scores from the pre-pandemic survey (2019) to the survey undertaken during the pandemic (2021). Since COVID-19 was a period of many changes and uncertainty, students may have found some comfort in continuing their academics from home, which could have contributed to decreases in stress levels. The results of the HIT-6 scores showed an improvement from 2019 to 2021, suggesting that the frequency of headaches may have shown a downward trend after COVID-19. This trend is similar to the results obtained by Parodi et al. (2020), who also indicated a decrease in the frequency and intensity of migraine episodes during the COVID-19 lockdown. A reduction in these symptoms could also be attributed to less exposure to stressors such as loud noises, social interactions, and reduced physical stress, along with better quality of sleep. It may also be due to the combined results of cutting down on taxing social lives, studying from home, and the ability to organize and be flexible with one’s curriculum and work.

Most of our results did not show any statistically significant differences for the COVID-19 outbreak and lockdowns on the HIT-6 and PSS-10 scores or on the frequency, intensity, and duration of migraines. Our results showed that 53.4% of the students who had COVID-19 infections reported no influence on their migraine-like headaches. These findings suggest possible decreases in the stress levels and frequency and severity of migraine-like headaches in students who had COVID-19. With the lockdowns, school-related worries were less, which may have affected the impact of these headaches. Along with this, social overloads decreased, which would have led to higher self-awareness, thus reducing exhaustion levels.

### Strengths and Limitations

This study is one of few that have looked at the effects of the COVID-19 pandemic on chronic illnesses such as migraines in a novel student population using validated instruments. The study occurred over two time periods: before (2019) and during (2021) the COVID-19 pandemic. Although our study was able to provide some inferences on the effect of the COVID-19 pandemic on migraine-like headaches in university students, our study is not without its limitations. The results obtained from this study cannot be directly generalized as only the student population was considered. Additionally, the data was collected from the students of only one university, and the response rate was low, which might affect the generalizability of the results [38]. Further, since personal connections could not be established due to lockdowns, the surveys were sent online, and this may have decreased the response rates for the questionnaires and had an overall impact on causal inference. The major drawback of this study is in the formatting of the online questionnaire wherein the participants self-diagnosed their migraines. Moreover, the survey did not specify whether the participants had a relative living in the same household with COVID-19 during the time of the surveys or had themselves been infected. In addition, we were not able to delve into the causes of the increased stress and poor scores on the HIT-6 scale. However, we assumed that these poor scores could be due to increased academic stressors, poor sleep quality, and other life stressors such as moving to a new city, job searches, and extracurricular activities. The unforeseen effects of the lockdowns could be a contributing factor to not being able to assess the various aspects of migraine stressors. Due to the different phases of the lockdowns (restricted and relaxed phases), the long-term effects of the lockdowns on migraine headaches could not be assessed in a true sense. This study did not separate migraines by type, and therefore should not be considered in future research.

## 5. Conclusions

Significant differences (*p* = 0.044) were found in the HIT-6 score <49 category, while the differences in the other categories were not significant. From 2019 to 2021, more students experienced lower impacts on their role functioning, indicating that university students were having less severe migraines. Similar trends were seen in the results for the PSS-10, showing lower stress levels in 2021 compared to 2019. Overall, our study depicted a direction, although not statistically significant, towards lower scores on the validated tools for quantitatively assessing stress and role functioning in university students with migraine-like headaches preceding the COVID-19 infection and lockdowns and subsequently following it. Despite our findings, specific educational programs are needed to raise awareness about migraines. Relaxation programs, stress management courses, and providing guidance to avoid stress are required. We suggest further large-scale longitudinal studies using standard clinical diagnostic tools to report on the worldwide prevalence and trigger factors associated with understanding the epidemiology of migraine-like headaches. Additionally, having a medically diagnosed migraine may improve the validity of future studies. Interesting results could be obtained by assessing mood in future studies.

## Figures and Tables

**Table 1 ijerph-20-05499-t001:** Response rates.

Year	Number of Respondents	Number of UT Enrollments	Response Rate (%)
2019	721	9535	7.6%
2021	520	9304	5.6%
Total	1247	18,839	6.6%

**Table 2 ijerph-20-05499-t002:** Demographics of the 2019 and 2021 samples.

Year	n	Mean	SD	Minimum	Q1 (25th Percentile)	Median	Q3 (75th Percentile)	Maximum
2019	721	20.81	4.32	17.00	18.00	20.00	21.00	48.00
2021	520	20.95	3.19	16.00	19.00	20.00	22.00	41.00
Overall	1241	20.87	3.89	16.00	19.00	20.00	21.00	48.00

**Table 3 ijerph-20-05499-t003:** Gender distribution.

Gender	2019		2021		Overall	
	Count	Percentage	Count	Percentage	Count	Percentage
Male	136	18.86	131	25.19	267	21.51
Female	578	80.17	384	73.85	962	77.52
Other	7	0.96	5	0.96	12	0.97
Total	721	100	520	100	1241	100

**Table 4 ijerph-20-05499-t004:** Categorical values of the PSS-10 and HIT-6 scores when comparing the 2019 and 2021 cohorts.

PSS-10 Score Categories	2019 (n)	2021 (n)	*p*-Values
27–40	422	316	0.444
14–26	212	140	0.299
0–13	2	5	0.231
HIT-6 Score Categories			
>60	317	198	0.373
56–59	69	40	0.514
50–55	51	38	0.514
<49	16	20	0.044 *

* *p* < 0.05.

## Data Availability

The data are available by request from the corresponding author.

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
