# Peer review of "Perceived Stress and Impact on Role Functioning in University Students with Migraine-Like Headaches during COVID-19"

_ijerph, 2023, doi:10.3390/ijerph20085499_

Round 1

Reviewer 1 Report

The article presented to me for review is on "Perceived Stress and Impact on Role Functioning in University Students with Migraine-like Headache during COVID-19" and compares parameters in 2019 and 2021 on rather large groups of students - 721 and 520, respectively.

However the paper is well written and contains interesting and even surprising conclusions attention some additions before it is accepted for publication: 

1. The abstract uses the phrase "Migraines, a chronic issue, can be debilitating.... ". Migraine is a disease not "an issue" and this should be corrected. 

2. I do not fully understand the use of the phrase "Migraine-like Headache". Were students with a medically confirmed diagnosis of migraine invited to participate in the study, or was it a group with headaches and the authors of the paper made the diagnosis based on the questionnaire? Even if it was the second option - having accurate criteria for the diagnosis of migraine ICHD-3 can be used to select patients with migraine. What type of headache is "Migraine-like Headache"? Trigeminal-autonomic, suboccipital neuralgia? this should be specified in the method and if the authors do not have such knowledge it should be written about in the limitations of the work. 

3. It would be worthwhile in the introduction or in the discussion to add a little information about the potential impact of the virus itself on the potential pathogenesis of headaches based on: https://pubmed.ncbi.nlm.nih.gov/35758225/

Author Response

Thank you for your comments. the responses are below: 

  1. This has been changed accordingly on line 10. This is a good point that migraines are not an issue, rather they are a disease.
  2. Participants in the study did not receive a medical diagnosis of confirmed migraine. Rather, as it is mentioned throughout the methods, this was a self-reported survey-based study that collected data on migraines based of the student’s subjective opinion. We agree that one of the methods to make this study better would have been to include the ICHD-3, however we did not think of that at the time of creating the survey, and we have listed this as a limitation in the strength and limitations section. Additionally, we did not specify the exact type of migraine that is suffered by the participants (i.e., trigeminal-autonomic and suboccipital neuralgia). We reported this in the methods section, and we addressed this limitation in the limitations paragraph.
  3. Agreed, we added this reference that you have suggested to line 44 with additional information as well. Although you have mentioned to discuss the pathogenesis of headache in relation COVID, we have discussed this throughout the introduction.

Reviewer 2 Report

Interestin paper but the partecipants self-diagnoses of migraine does not allow the correct medical diagnose of headache so we can't assure the clinical and demographycal features of episodic/chronic migraine or tension type headache. Partecipants mood condition with validated assessment at different timepoints (2019 and 2021) would have been helpful to better understand the real impact of COVID-19 pandemia.

Author Response

Thank you for your comments. 

  1. We agree that because this was self-report/self-diagnosis, we cannot medially diagnosis the headache/migraine of participants. Additionally, that was not our intention in the study and we have mentioned this limitation in the paper. Also, we did utilize validated tools (e.g., PSS and HIT-6) at the different time points (i.e., 2019 and 2021). These tools do not relate to COVID itself, however it is still relevant and were valid measurements.

Reviewer 3 Report

The aim of this study 11 is to identify the impact of COVID-19 on the role functioning and perceived stress levels 12 of students suffering from migraine-like headaches, and the topic is very interesting and can be included in a series of small studies with the same purpose.

The objective of the study was not fully achieved, as the statistical impact for the main indicator is barely significant, with a p=0.044, that limits the clinical significance of this data.

On the other hand, the authors correctly underline the limits of the study, from the number to the difficulty of sampling, the arbitrariness of the diagnosis of migraines, which is probably the major limitation of the study, to the fact that it is monocentric, etc.

Minor concerns are:

Line 22 Please use “suggesting” instead indicating

In the last sentence the conclusions can be shared but it should be added that the data obtained from this study cannot be directly generalized.

Author Response

Thank you for your comments. 

  1. After reading your comment regarding the objective of our study, we believe as a group that our objective was achieved. In fact, we discovered findings that are surprising and novel. While doing our statistics, we set our p value to 0.05. The fact that we got a p value of 0.044 indicates that we have reached statistical significance. Yes, indeed it does limit the clinical significance of these findings not only because of our p value, but the fact that we didn’t have the participants be clinically diagnosed. This is mentioned in the limitations of our paper.
  2. For line 22, we have fixed this and added suggesting instead of indicating.
  3. We agree with this comment, this has been changed accordingly and you can see the changes in lines 199-203.